# Epigenetic Age Acceleration in Frontotemporal Lobar Degeneration: A Comprehensive Analysis in the Blood and Brain

**DOI:** 10.3390/cells12141922

**Published:** 2023-07-24

**Authors:** Megha Murthy, Patrizia Rizzu, Peter Heutink, Jonathan Mill, Tammaryn Lashley, Conceição Bettencourt

**Affiliations:** 1Queen Square Brain Bank for Neurological Disorders, UCL Queen Square Institute of Neurology, London WC1N 1PJ, UKt.lashley@ucl.ac.uk (T.L.); 2Department of Clinical and Movement Neurosciences, UCL Queen Square Institute of Neurology, London WC1N 1PJ, UK; 3German Center for Neurodegenerative Diseases (DZNE), 72076 Tübingen, Germany; 4Alector, Inc., South San Francisco, CA 94080, USA; 5Department of Clinical and Biomedical Sciences, Faculty of Health and Life Sciences, University of Exeter, Exeter EX4 5DW, UK; 6Department of Neurodegenerative Disease, UCL Queen Square Institute of Neurology, London WC1N 1PJ, UK

**Keywords:** frontotemporal lobar degeneration, frontotemporal dementia, progressive supranuclear palsy, DNA methylation aging, epigenetic clock

## Abstract

Frontotemporal lobar degeneration (FTLD) includes a heterogeneous group of disorders pathologically characterized by the degeneration of the frontal and temporal lobes. In addition to major genetic contributors of FTLD such as mutations in *MAPT*, *GRN*, and *C9orf72*, recent work has identified several epigenetic modifications including significant differential DNA methylation in *DLX1*, and *OTUD4* loci. As aging remains one of the major risk factors for FTLD, we investigated the presence of accelerated epigenetic aging in FTLD compared to controls. We calculated epigenetic age in both peripheral blood and brain tissues of multiple FTLD subtypes using several DNA methylation clocks, i.e., DNAmClock_Multi_, DNAmClock_Hannum_, DNAmClock_Cortical_, GrimAge, and PhenoAge, and determined age acceleration and its association with different cellular proportions and clinical traits. Significant epigenetic age acceleration was observed in the peripheral blood of both frontotemporal dementia (FTD) and progressive supranuclear palsy (PSP) patients compared to controls with DNAmClock_Hannum_, even after accounting for confounding factors. A similar trend was observed with both DNAmClock_Multi_ and DNAmClock_Cortical_ in post-mortem frontal cortex tissue of PSP patients and in FTLD cases harboring *GRN* mutations. Our findings support that increased epigenetic age acceleration in the peripheral blood could be an indicator for PSP and to a smaller extent, FTD.

## 1. Introduction

Frontotemporal lobar degeneration (FTLD) refers to a heterogeneous group of disorders that are pathologically characterized by the degeneration of the frontal and temporal lobes resulting in clinical manifestations that predominantly include a progressive decline in behavior or language [1,2]. FTLD is the third most common cause of dementia (termed frontotemporal dementia (FTD)) following Alzheimer’s disease (AD) and Dementia with Lewy Bodies [3]. Patients presenting with dementia due to FTLD can typically be grouped into one of three clinical categories based on their early and predominant symptoms: behavioral variant frontotemporal dementia (bvFTD), and two language variants, semantic dementia (SD), and primary progressive non-fluent aphasia (PNFA) [3]. In addition, FTLD encompasses a spectrum of other neurodegenerative clinical phenotypes, including atypical forms that overlap with motor neuron disease/amyotrophic lateral sclerosis (MND/ALS) (FTD-MND, FTD-ALS) and with atypical parkinsonian disorders such as corticobasal degeneration (CBD), and progressive supranuclear palsy (PSP) [1,4]. Based on the neuropathology and nature of the proteinaceous aggregates, FTLD can be characterized mainly into, FTLD with inclusions of hyperphosphorylated tau (FTLD-tau); FTLD with ubiquitin immunoreactive neuronal inclusions, which include the 43 kDa transactive response DNA-binding protein (TDP-43) inclusions (FTLD-TDP), the fused in sarcoma (FUS) inclusions (FTLD-FUS), and the unidentified ubiquitin-positive inclusions (FTLD-UPS); and a small population of FTLD with no inclusions (FTLD-ni) [1,5].

Familial forms of FTLD account for up to 30–40% of all cases, with mutations in *MAPT*, *GRN*, and *C9orf72* accounting for a majority of the cases [4]. Recent studies have also reported several epigenetic modifications in various FTLD subtypes including significant differential methylation in the 17q21.31 locus (which includes *MAPT*) in the peripheral blood of individuals with PSP, and to a lesser extent in FTD [6]; hypermethylation in *DLX1* was also reported in the prefrontal cortex of individuals with PSP [7]. A recent study by Fodder et al. also conducted a meta-analysis and identified that hypomethylation in *OTUD4* was associated with FTLD across pathological subgroups and subtypes [8].

One of the major risk factors for most complex neurodegenerative disorders and dementia is aging. FTD is a predominantly early onset form of dementia, typically seen in individuals under the age of 65 years, and although only 20–25% of the cases present in old age, aging remains to be one of the biggest risk factors [9]. Therefore, in addition to deciphering the complex etiology and molecular mechanisms that occur due to the heterogeneity brought about by the genetic, epigenetic, and environmental factors, it is also important to address the effect of aging in the development and progression of diseases within the FTLD spectrum. However, the association of biological markers of aging with risk of this disease spectrum remains largely unexplored. 

Epigenetic clocks have proven to be excellent estimators of biological age and have been repeatedly used as biomarkers of biological age. Epigenetic clocks are DNA methylation (DNAm) based biomarkers that use penalized regression models, such as elastic net regression to select a subset of DNAm sites that can be used to estimate the DNAm age of any tissue or cell type [10]. Epigenetic age acceleration can then be calculated by comparing the difference between DNAm age and chronological age, wherein a positive age acceleration value indicates that the tissue is biologically older than expected and vice versa for a negative age acceleration value. The first pan-tissue epigenetic clock was created by Horvath, followed by multiple other tissue specific clock such as the blood tissue specific clock created by Hannum et al., and a more recent cortical tissue specific clock created by Shireby et al. [11,12,13]. These first-generation predictors of age were followed by several other epigenetic clocks called the second-generation clocks, which were developed as predictors of lifespan and health (PhenoAge), mortality (GrimAge), and clocks which showed strong associations with other phenotypic traits [14,15]. Accelerated epigenetic aging has been shown to be associated with various clinical traits, disease phenotypes, as well as altered cellular proportions in several tissues and disease contexts including AD and other neurodegenerative diseases [16,17]. A previous study in *C9orf72* mutation carriers with FTD, FTD-ALS, and ALS clinical phenotypes identified that an increase in DNAm age acceleration was associated with an earlier age of onset and shorter disease duration in the blood, but with just earlier onset in the frontal cortex and spinal cord tissue, thus reflecting the severity/progression of the disease [18]. 

Therefore, with the aim of comprehensively investigating the role of these DNAm based biomarkers of aging in the different subtypes of FTLD, in both peripheral blood and post-mortem brain tissue, we performed DNA methylation-based clock analyses using multiple blood, cortical, and pan-tissue epigenetic clocks.

## 2. Materials and Methods

### 2.1. Study Overview/Design

#### 2.1.1. Peripheral Blood Samples

Cohort 1 comprised publicly available peripheral blood epigenome-wide DNA methylation profiles of FTD, PSP, and control individuals (Gene Expression Omnibus—GEO accession number GSE53740) [6]. The dataset originally consisted of DNA methylation profiles generated from the peripheral blood of patients with neurodegenerative disorders (*n* = 190; 121 FTD, 7 FTD-MND, 43 PSP, and 15 AD, 1 CBD, and 4 with unknown diagnosis), and healthy controls (*n* = 193) enrolled as part of a large genetic study in neurodegenerative dementia (Genetic Investigation in Frontotemporal Dementia, GIFT) at the UCSF Memory and Aging Center (UCSF-MAC) [19]. The FTD-MND cases were merged with FTD group, and the CBD case was merged with the PSP group. AD samples were not included in our study as the analysis focused on diseases under the FTLD umbrella; 4 samples with unknown diagnosis were also excluded.

#### 2.1.2. Post-Mortem Brain Tissue Samples

Cohort 2 consisted of epigenome-wide DNA methylation profiles generated from the frontal cortex grey matter of post-mortem brain tissues from 16 individuals with FTD with TDP-43 pathology (i.e., FTLD-TDP type A (TDPA, *C9orf72* mutation carriers), (*n* = 8); and FTLD-TDP type C (TDPC, sporadic), (*n* = 8)), and 8 neurologically normal controls. All post-mortem brain tissues in cohort 2 were donated to the Queen Square Brain Bank archives and are stored under a license from the Human Tissue authority (No. 12198) as described by Fodder et al. [8]. Cohort 3 consisted of epigenome-wide DNA methylation profiles generated from the frontal lobe of post-mortem brain tissues from 33 individuals with FTLD (FTLD-TDP types A and B (*GRN* (*n* = 7) and *C9orf72* (C9, *n* = 13) mutation carriers, respectively), and FTLD-tau (FTDP-17—*MAPT* mutation carriers, *n* = 13)) and 14 neurologically normal controls. All post-mortem tissues in cohort 3 were obtained under a Material Transfer Agreement from the Netherlands Brain Bank, and MRC King College London, as described by Menden et al. [20]. Cohort 4 consisted of epigenome-wide DNA methylation profiles generated from post-mortem prefrontal lobe of 94 individuals with PSP and 72 controls for which data were made publicly available (GEO accession number GSE75704) [7].

### 2.2. DNA Methylation Data Pre-Processing

For cohorts 1 and 4, DNA methylation profiling was performed using the Infinium Human Methylation450 BeadChip (Illumina, San Diego, CA, USA), as described by Li et al. and Weber et al., respectively [6,7]. DNA methylation profiling for cohorts 2 and 3 were performed using the Infinium HumanMethylationEPIC BeadChip (Illumina, San Diego, CA, USA). Sample processing steps and detailed methodology have been described previously [8,20]. Raw files (methylated and unmethylated intensity files in case of cohort 1 and .idat flies for cohorts 2–4) for the DNA methylation profiles for all cohorts were subjected to harmonized quality control and pre-processing steps using ChAMP (v. 2.21.1), minfi (v.1.46.0), and WateRmelon (v.2.6.0) R (v.4.2.1) packages as previously described [21,22,23,24]. Briefly, raw intensity files were subjected to rigorous quality control checks, which included filtering out failed and atypical samples as well as outlier detection. This was followed by the removal of samples with <80% bisulfite conversion using bisulfite conversion assessment, which converts probe intensities into percentage. Poorly performing probes were filtered out if they had a bead count of <3 in more than 5% of the samples, or if over 1% of samples showed a detection *p*-value > 0.05. In addition, samples were excluded if they showed >1% probes above the 0.05 detection *p*-value threshold, and if sex predictions did not match with phenotypic sex. 

More detailed characterization of the samples that passed the aforementioned quality control checks in each cohort is given in Table 1. Dasen normalization was carried out for all cohorts except Cohort 1, for which quantile normalization performed (as .idat files were unavailable). Cell proportions in the blood were estimated from DNA methylation data using methods implemented into the advanced analysis option of the epigenetic age calculator software, which employs both Houseman’s method to estimate proportions of CD8 T cells, CD4T cells, natural killer cells, B cells, monocytes and granulocytes [22,25], and Horvath’s method to estimate abundance measures of plasma blasts, CD8 + CD28-CD45RA- T cells, naive CD8 T cells, and naive CD4 T cells [12]. Cell proportion estimations in the bulk brain tissues were performed to classify cell types into neuron-enriched (NeuN+), oligodendrocyte-enriched (SOX10+), and other brain cell types (NeuN-/SOX10-) populations using the CETYGO R package (https://github.com/ds420/CETYGO (accessed on 17 February 2023)) as previously described [26,27]. 

### 2.3. Epigenetic Clocks and Estimations of DNAm Age Acceleration 

For the peripheral blood dataset (Cohort 1), DNAm age estimation was performed using 4 clocks designed either for pan-tissues or specifically designed for blood, namely DNAmClock_Multi_ [12], DNAmClock_Hannum_ [11], PhenoAge [14], and GrimAge [15]. For the post-mortem brain datasets (Cohorts 2–4), DNAm age estimation was performed using 2 clocks, the pan-tissue DNAmClock_Multi_ and the cortical tissue specific DNAmClock_Cortical_ [13]. DNAm ages for DNAmClock_Multi,_ DNAmClock_Hannum_, PhenoAge, and GrimAge were calculated using the advanced analysis with normalization, on the online calculator (http://dnamage.genetics.ucla.edu/ (accessed on 03 December 2022)). In addition to the DNAm age estimates and age acceleration residuals, the online calculator also provided Intrinsic Epigenetic Age Acceleration (IEAA) measures for the DNAmClock_Multi_ and DNAmClock_Hannum_, as well as Extrinsic Epigenetic Age Acceleration (EEAA) measures. IEAA is the residual obtained from a multivariable regression of DNAm age on chronological age and blood cell count estimates; and is therefore unaffected by both variation in chronologic age and blood cell composition, making it a measure of cell-intrinsic aging [28]. EEAA, on the other hand, are residuals that are obtained by combining Hannum DNAm age with three blood cell components (naïve cytotoxic T cells, exhausted cytotoxic T cells, and plasmablasts) to form an aggregate measure (enhanced Hannum DNAm age followed by regression onto chronological age) [29]. EEAA is a measure that is dependent on age-related changes in the blood cell composition and integrates known age-related changes in blood cell counts with a blood-based measure of epigenetic age before adjusting for chronologic age and therefore is a measure of immune system aging [30]. The DNAm ages for DNAmClock_Cortical_ were calculated as described by Shireby et al. [13]. For all clocks, standard linear regression models were applied and DNAm age acceleration was calculated as the residual obtained by linear regression of DNAm age on chronological age and adjusting for possible confounders such as tissue-specific cell type proportions. A brief overview of the study design is described in Figure 1; the number of individuals included in each cohort represent the samples remaining after quality control (Figure 1).

### 2.4. Statistical Analysis

Comparisons for statistical significance in the DNAm age acceleration between cases and controls across groups/brain regions were performed using Kruskal–Wallis test, and pairwise comparisons between groups (i.e., FTLD subtypes vs. controls) within each cohort were performed using pairwise Wilcoxon test with the Benjamini–Hochberg multiple testing correction. Correlations between epigenetic age acceleration (i.e., residuals obtained by linear regression of DNAm age on chronological age) and cell-type proportions (e.g., estimates of neuronal proportions) and/or disease traits (e.g., disease onset) were calculated using Pearson’s coefficient.

## 3. Results

### 3.1. Correlation between DNAm Age and Chronological Age in the Peripheral Blood and Post-Mortem Brain Tissue Cohorts

For the peripheral blood dataset (Cohort 1), significant strong correlations were observed between chronological age and DNAm age in all 4 clocks, DNAmClock_Multi_ (r = 0.79, *p* = 1 × 10^−75^), DNAmClock_Hannum_ (r = 0.81, *p* = 2 × 10^−82^), PhenoAge (r = 0.71, *p* = 8.5 × 10^−55^), and GrimAge (r = 0.88, *p* = 3.1 × 10^−114^), with GrimAge showing the strongest correlation, highest significance, and lowest error (defined as median absolute deviation between DNAm age and chronological age) (Appendix A). Similar to that observed in the peripheral blood, significant strong correlations were also observed between chronological age and DNAm age for both DNAmClock_Multi_ (r = 0.7–0.94, *p* = 5.4 × 10^−5^–1.8 × 10^−25^) and DNAmClock_Cortical_ (r = 0.81–0.97, *p* = 6.9 × 10^−8^–2.2 × 10^−39^) in the post-mortem brain samples, with Cohort 3 (FTLD-TDPB *C9orf72*, FTLD-TDPA *GRN*, and FTLD-Tau *MAPT* mutation carriers) showing the strongest correlation and highest significance with both clocks (Appendix A). Overall, DNAmClock_Cortical_ predominantly displayed stronger correlations with higher significance in all post-mortem brain tissue cohorts. The median absolute deviation (error) varied between the cohorts and the clocks, ranging between 4.1 and 15, with the lowest for DNAmClock_Multi_ and highest for DNAmClock_Cortical_ in Cohort 2. An underestimation of DNAm age compared to actual chronological age was observed with DNAmClock_Multi_ in all brain tissue cohorts except Cohort 2, whereas DNAmClock_Cortical_ generally tended towards DNAm age overestimations (Appendix A).

### 3.2. Epigenetic Age Acceleration in the Peripheral Blood of Individuals with a Clinical Diagnosis of FTD and PSP

Significant epigenetic age acceleration was observed for the FTD (AgeAccel = ~2 years, *p* = 0.002) and PSP cases (AgeAccel = ~4 years, *p* = 0.0006) compared to controls with DNAmClock_Hannum_ (Figure 2b and Appendix A), with similar trends (although not statistically significant) observed with DNAmClock_Multi_ (Figure 2a). Age acceleration remained significant upon adjustment for differences in blood cell counts, as observed by IEAA_Hannum_ for both FTD (AgeAccel = ~2 years, *p* = 0.03) and PSP (AgeAccel = ~3 years, *p* = 0.01) when compared to controls, and a similar result was observed with IEAA_Multi_ (Figure 2e,f). Further, significant age acceleration was observed with EEAA, which accounts for known age-related changes in blood cell counts during epigenetic age estimation and is a measure of immune system aging, for both FTD (AgeAccel = ~3 years, *p* = 0.0003) and PSP (AgeAccel = ~5 years, *p* = 0.0003), when compared to controls (Figure 2g). A trend in age acceleration (~2 years) for FTD only was observed with the PhenoAge epigenetic estimates, whereas with the GrimAge clock, age acceleration was observed in both FTD (~1.5 years) and PSP (~1.5 years), compared to controls, although no statistical significance was observed upon pairwise comparison (Figure 2c,d).

### 3.3. Epigenetic Age Acceleration in Post-Mortem Brain Tissue of Pathologically Confirmed FTLD Subtypes

Similar to what was observed in blood, in the brain tissue of sporadic PSP cases (Cohort 4) a trend towards epigenetic age acceleration (~1 year) was observed with both DNAmClock_Multi_ and DNAmClock_Cortical_ compared to controls (Figure 3c,f). Epigenetic age acceleration, albeit not statistically significant, was also observed for the FTD-TDPA *GRN* mutation carriers (Cohort 3) with DNAmClock_Multi_, with concordant results from the DNAmClock_Cortical_ (Figure 3b,f). The *C9orf72* mutation carriers of both FTLD-TDP types A (Cohort 2) and B (Cohort 3), as well as the sporadic FTLD-TDP subtype C (Cohort 2) and the *MAPT* mutation carriers (Cohort 3), however, showed no consistent evidence in favor of age acceleration compared to controls with both DNAmClock_Multi_ and DNAmClock_Cortical_.

Previous studies, including ours have revealed that cell-type composition in a specific tissue influences DNAm age estimation and thus epigenetic age acceleration [13,26]. Therefore, we analyzed the association of epigenetic age acceleration in the post-mortem brain tissues using neuronal and oligodendrocyte proportion estimates obtained from the cell-type deconvolution algorithm CETYGO [27]. In agreement with the previous reports, a significant negative correlation was observed between age acceleration and neuronal proportions for both control and disease groups in all cohorts with DNAmClock_Cortical_; a concordant result was also observed with DNAmClock_Multi_ in cohort 4 (Figure 4). Positive correlations were observed between oligodendrocyte proportions and epigenetic age acceleration for all cohorts with both clocks except for Cohort 2 with DNAmClock_Cortical_, which could be due to the fact that unlike the other brain tissue cohorts, this cohort comprises grey matter instead of a mix of grey and white matter, and thus lower proportions of oligodendrocytes (Figure 4).

### 3.4. Association of Epigenetic Age Acceleration with Disease Onset and Duration 

A previous study reported significant inverse associations between age acceleration (defined in that case as the difference between DNAm age and chronological age) and clinical traits such as disease onset and duration in the blood of patients with *C9orf72* repeat expansions (ALS, ALS-FTD, and FTD), with a similar trend observed in the spinal cord, frontal and temporal cortices in ALS and ALS-FTD patients with disease onset, but not with disease duration [18]. In our brain datasets, for the FTLD-TDPA *C9orf72* mutation carriers in Cohort 2, weak negative correlations were observed between age acceleration (residuals) and disease onset with DNAmClock_Multi_, but not with DNAmClock_Cortical_ (Appendix A); no negative correlation was observed with disease onset for FTLD-TDPB *C9orf72* mutation carriers in Cohort 3 with both clocks (Appendix A). As Zhang et. al. [18] found a trend towards inverse associations between age acceleration difference, we also examined the associations between age acceleration difference and disease onset and results were very to those we observed for the residuals (Appendix A). Zhang et al. [18] also reported inverse associations between age acceleration and disease duration in blood and in the temporal cortex; in line with that, a significant inverse association between age acceleration (both difference and residuals) and disease duration was observed in individuals with *C9orf72* mutations of FTLD-TDPB subtype in Cohort 3 with DNAmClock_Multi_ (Figure 5c,g), and a similar trend was observed with DNAmClock_Cortical_ (Figure 5d,h). Individuals with *C9orf72* mutations of FTLD-TDPA subtype in Cohort 2 also showed a trend towards inverse correlations with disease duration with both clocks (Figure 5a,b,e,f). *GRN* mutation carriers of FTLD-TDPA subtype also showed significant strong inverse correlation between age acceleration (residuals) and disease onset with DNAmClock_Cortical_, whereas *MAPT* mutation carriers showed significant strong inverse correlation between age acceleration (residuals) and disease onset with DNAmClock_Multi_ (Appendix A).

## 4. Discussion

Accelerated aging has been shown to be an important predictor of several age-related diseases including cancer [31], diabetes [32], as well as neurodegenerative diseases such as AD [33]. Moreover, associations between accelerated epigenetic age and various clinical traits, phenotypes, and cellular proportions have also been reported [17,34]. Our study aimed to systematically evaluate the presence of accelerated epigenetic aging in multiple neurodegenerative conditions occurring as a result of FTLD in the peripheral blood using multi-tissue and blood specific epigenetic clocks such as DNAmClock_Multi_, DNAmClock_Hannum_, PhenoAge, and GrimAge, as well as in the case of post-mortem brain tissue using DNAmClock_Multi_ and the brain tissue specific DNAmClock_Cortical_. Our analysis of epigenetic age acceleration in the peripheral blood revealed significant age acceleration in both FTD and PSP individuals compared to controls with DNAmClock_Hannum_, with a concordant trend being observed with DNAmClock_Multi_. These results remained significant even after accounting for differences in blood cell counts (IEAA) and upon accounting for known age-related changes in blood cell counts during epigenetic age estimation (EEAA) (Figure 2, Appendix A). A similar trend was observed in the post-mortem brains, with epigenetic age acceleration being observed in PSP patients compared to controls with both DNAmClock_Multi_ and DNAmClock_Cortical_ (Figure 3c,f) and a trend towards epigenetic age acceleration in the *GRN* mutation carriers (FTLD-TDPA) (Figure 3b,e).

For the peripheral blood cohort, comparing the blood and multi-tissue clocks, DNAmClock_Hannum_ showed stronger correlation, higher significance, and lower error values compared to DNAmClock_Multi_, as expected. For the brain tissue cohorts similarly, as expected, DNAmClock_Cortical_ showed higher correlations with stronger significance between DNAm age and chronological age for all cohorts; however, the error values were not necessarily lower compared to DNAmClock_Multi_. Specifically, strong correlations between DNAm age and chronological age observed with Cohort 2 demonstrate the applicability of both clocks to grey matter tissues in addition to a mix of white and grey matter (Cohorts 3 and 4) as well as to white matter tissues, as previously demonstrated by our group [26]. In addition, concordant to previous reports, age acceleration was generally positively correlated with oligodendrocyte proportions and negatively correlated with neuronal proportions in brain tissue indicating the role of cellular proportions, which are typically altered in disease, towards epigenetic age estimations and validating our previous findings [26].

Epigenetic age acceleration observed with DNAmClock_Hannum_ in the peripheral blood (Cohort 1) in both FTD and to a larger extent in PSP compared to controls remained significant even after accounting for differences in the blood cellular composition (IEAA_Hannum_). Neuroinflammation has been shown to be a major component in the pathology and progression of several neurodegenerative diseases including FTD [35]. Dysregulation of the peripheral immune system has also been previously reported, with an increased expression observed in genes associated with adaptive immune cells (CD19+ B-cells, CD4+ T-cells, and CD8+ T-cells) and decreased expression in genes associated with innate immune cells (CD33^+^ myeloid cells, CD14^+^ monocytes, BDCA4^+^ dendritic cells, and CD56+ natural killer cells) in FTD participants compared to healthy aging [36]. These differences in the peripheral immune system in FTD compared to healthy aging makes it crucial that we account for known age-related changes in blood cell counts (EEAA); epigenetic age acceleration remained significant even after accounting for these changes in both FTD and PSP supporting the significant increase in epigenetic age of the immune system in FTLD compared to controls. Similar trends in epigenetic age acceleration were also observed with DNAmClock_Multi_ and IEAA_Multi_. These findings strongly suggest that increased epigenetic age in the peripheral blood can be an indicator for PSP and, to a smaller extent, FTD [35,36].

A similar trend in accelerated epigenetic aging could be observed in post-mortem brains of PSP patients compared to controls with both DNAmClock_Multi_ and DNAmClock_Cortical_; however, for the FTD subtypes, only the *GRN* mutation carriers (FTLD-TDPA) showed a consistent trend towards epigenetic age acceleration. The concordance in age acceleration patterns in the blood (~4 years) and brain (~1 year) in case of PSP could be an indicator of shared methylation patterns and shared systemic aging related processes occurring in both tissues. Epigenetic age acceleration has also been reported in other neurodegenerative diseases, such as in the blood of Parkinson’s disease (PD) patients compared to controls, where increased age acceleration with IEAA_Multi_ and EEAA were observed [37], and in a longitudinal study of control individuals that revealed increased DNAm age in the blood to be a significant predictor of dementia at follow-up after 15 years [38]. 

Findings from a previous report also correlated epigenetic age acceleration difference measures in ALS/FTD patients with *C9orf72* mutations with a more severe disease phenotype as represented by shorter disease duration and earlier age of onset primarily in the blood, and to an extent, with an earlier age of onset in brain tissues [18]. However, this cohort consisted of only cases, and the lack of controls limited the study primarily to clinical phenotypes. In our datasets, we observed weak negative correlations between the age acceleration (both residuals as well as difference) and disease onset in the brains of FTLD-TDPA patients with *C9orf72* mutations with DNAmClock_Multi_; however, no negative correlation was observed for of FTLD-TDPB *C9orf72* mutation carriers in Cohort 3 with both clocks. Nevertheless, we did observe significant inverse association between age acceleration (both difference and residuals) and disease duration in individuals with *C9orf72* mutations of FTLD-TDPB subtype in Cohort 3 with DNAmClock_Multi_ (Figure 5), and a similar trend with DNAmClock_Cortical_. These results partially agree with the results of the previous study [18]; however, the sample sizes in both studies were relatively small and therefore these findings should be interpreted with caution. 

Our study has several limitations, including the fact that our blood and brain cohorts were not derived from the same individuals, and the brain cohorts 2 and 3 are relatively small. Further, we did not possess details regarding the clinical and neuropathological traits for Cohort 1, limiting the assessment of epigenetic age acceleration for different genetic and sporadic FTD subtypes or the association analysis with disease onset and duration in the peripheral blood dataset. Nevertheless, our study provides important groundwork by comparing epigenetic age acceleration measures for several FTLD phenotypes in both blood and brain tissues as well as their associations with clinical traits using multiple estimators of DNAm age. Future studies with larger sample sizes for each of the subtype, ideally investigating blood and brain tissue derived from the same individuals, are required to corroborate our findings. 

## 5. Conclusions

Our comprehensive analysis using several epigenetic clocks in both peripheral blood and post-mortem brain tissue cohorts reveals significant epigenetic age acceleration in the peripheral blood of individuals with FTD and PSP compared to controls as well as similar age acceleration trends in the brain tissue of individuals with PSP and *GRN* mutation carriers of FTLD-TDP type A.

## Figures and Tables

**Figure 1 cells-12-01922-f001:**
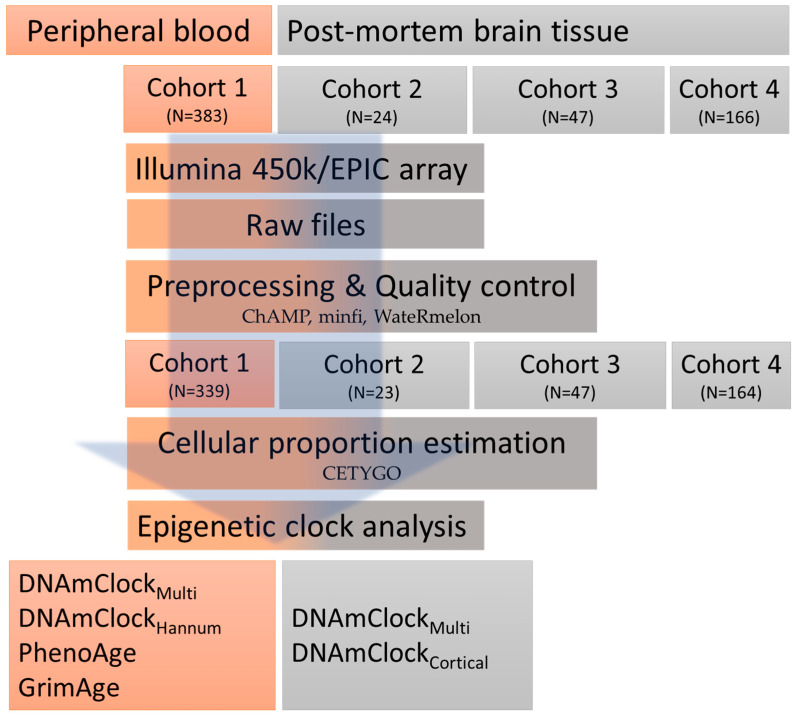
An overview of the study design.

**Figure 2 cells-12-01922-f002:**
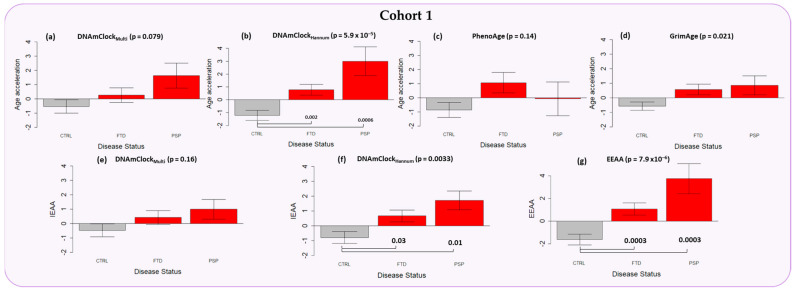
Epigenetic age acceleration in the peripheral blood samples of Cohort 1 (purple) constituting FTD (*n* = 117) and PSP cases (*n* = 44) as well as controls (*n* = 178) with the DNAmClock_Multi_, DNAmClock_Hannum_, PhenoAge, and GrimAge clocks. (**a**–**d**) Epigenetic age acceleration (y-axis) in relation to disease status (x-axis) with the 4 clocks; (**e**,**f**) intrinsic epigenetic age acceleration (IEAA, y-axis) of DNAmClock_Multi_ and DNAmClock_Hannum_ with respect to disease status (x-axis); and (**g**) extrinsic epigenetic age acceleration (EEAA, y-axis) with respect to disease status (x-axis). CTRL—control, FTD—frontotemporal dementia, PSP—progressive supranuclear palsy. Age acceleration residuals were obtained by regressing DNA methylation age against chronological age and adjusting for confounding factors such as cell type proportions. The bar plots depict the mean value and standard error (y-axis). *p*-values for across group comparisons were calculated using the Kruskal–Wallis test (*p*-values shown at the top of the plots (**a**–**g**)), and *p*-values for pairwise analysis between each disease group and controls were calculated using the Wilcoxon’s test with Benjamini–Hochberg correction for multiple testing (*p*-values shown at the bottom of the plots (**a**–**g**)).

**Figure 3 cells-12-01922-f003:**
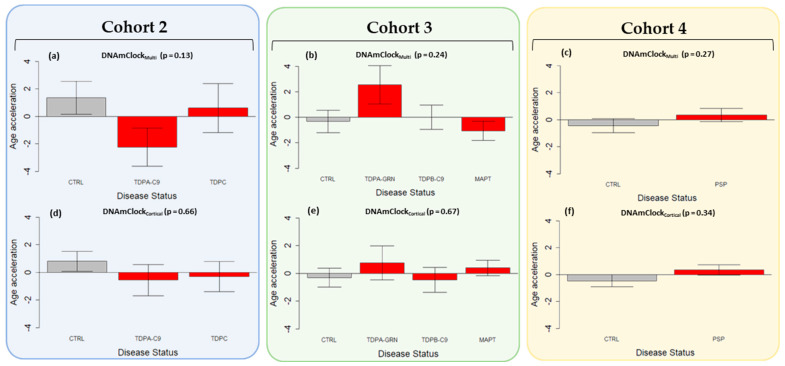
Epigenetic age acceleration in the post-mortem brain tissues using DNAmClock_Multi_ and DNAmClock_Cortical_. Cohort 2 (blue) constituted FTLD-TDP types A (*C9orf72* mutation carriers, *n* = 7), and C (sporadic cases, *n* = 8), and controls (*n* = 8), Cohort 3 (green) constituted FTLD-TDP types B (*C9orf72* mutation carriers *n* = 13), and A (*GRN* mutation carriers, *n* = 7), FTLD-Tau *MAPT* mutation carriers (*n* = 13) and controls (*n* = 14), and Cohort 4 (yellow) comprised PSP cases (*n* = 93) and controls (*n* = 71). (**a**–**f**) Epigenetic age acceleration (y-axis) in relation to disease status (x-axis) for the cohorts with DNAmClock_Multi_ and DNAmClock_Cortical_. CTRL—control, TDPA-C9—FTLD-TDPA (*C9orf72* mutation carriers), TDPC—FTLD-TDPC (sporadic); TDPA-GRN—FTLD-TDPA (*GRN* mutation carriers), TDPB-C9—FTLD-TDPB (*C9orf72* mutation carriers), MAPT—FTLD-Tau *MAPT* mutation carriers, PSP—progressive supranuclear palsy. Age acceleration residuals were obtained by regressing DNA methylation age against chronological age and adjusting for neuronal proportions; the bar plots depict the mean value and standard error (y-axis); *p*-values for across group comparisons were calculated using the Kruskal–Wallis test (**a**–**f**).

**Figure 4 cells-12-01922-f004:**
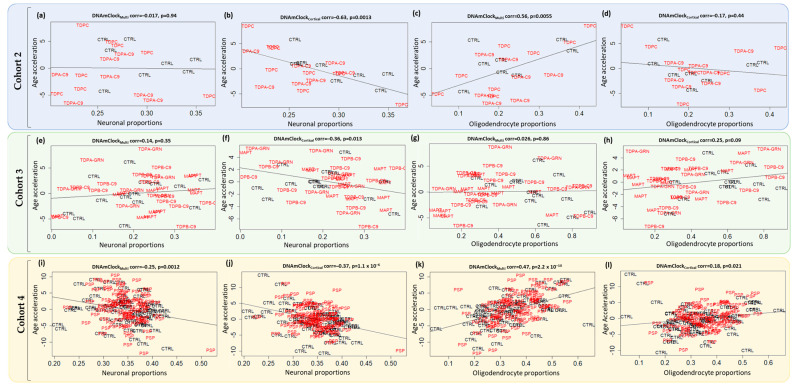
Associations between epigenetic age acceleration and cellular (neuronal and oligodendrocyte) proportions for DNAmClock_Multi_ and DNAmClock_Cortical_ in the different brain tissue cohorts. (**a**–**l**) Age acceleration residuals (y-axis) versus neuronal (NeuN positive) and oligodendrocyte (SOX10 positive) proportions (x-axis) for DNAmClock_Multi_ and DNAmClock_Cortical_ for Cohort 2 (blue; FTLD-TDPA *C9orf72*/FTLD-TDPC) (**a**–**d**), Cohort 3 (green; FTLD-TDPB *C9orf72*, FTLD-TDPA *GRN*, and FTLD-Tau *MAPT* mutation carriers) (**e**–**h**), and Cohort 4 (yellow; PSP) (**i**–**l**). Age acceleration residuals were obtained by regressing DNA methylation age against chronological age; cellular proportions were obtained using a DNA methylation-based cell-type deconvolution algorithm as described by Shireby et al. [27]. The correlation coefficient and *p*-values shown were calculated using Pearson correlation. CTRL—control, TDPA-C9—FTLD-TDPA (*C9orf72* mutation carriers), TDPC—FTLD-TDPC (sporadic); TDPA-GRN—FTLD-TDPA (*GRN* mutation carriers), TDPB-C9—FTLD-TDPB (*C9orf72* mutation carriers), MAPT—FTLD-Tau *MAPT* mutation carriers, PSP—progressive supranuclear palsy.

**Figure 5 cells-12-01922-f005:**
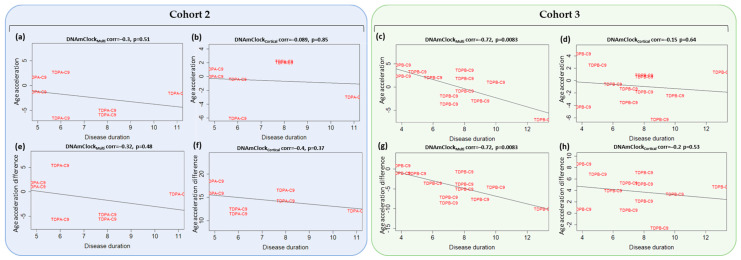
Association between age acceleration and duration with DNAmClock_Multi_ and DNAmClock_Cortical_ for the *C9orf72* mutation carriers in cohorts 2 (blue) and 3 (green). Age acceleration residuals (y-axis) for DNAmClock_Multi_ and DNAmClock_Cortical_ versus disease duration (x-axis) for Cohort 2 (**a**,**b**), and Cohort 3 (**c**,**d**), age acceleration difference (y-axis) versus disease onset (x-axis) for Cohort 2 (**e**,**f**) and Cohort 3 (**g**,**h**). Age acceleration residuals were obtained by regressing DNA methylation age against chronological age and adjusting for confounding factors such as neuronal proportions obtained using a DNA methylation-based cell-type deconvolution algorithm as described by Shireby et al. [27]. Age acceleration difference was the difference between DNA methylation age and chronological age. The correlation coefficient and *p*-values shown were calculated using Pearson correlation. TDPA-C9—FTLD-TDPA (*C9orf72* mutation carriers), TDPB-C9—FTLD-TDPB (*C9orf72* mutation carriers).

**Table 1 cells-12-01922-t001:** Cohort demographics.

**Peripheral Blood**			
**Sample Group**	**No. of Individuals**	**Females** (**%**) **[% Unknowns]**	**Average Chronological Age** (**SD**)
**Cohort 1**			
Controls	178	53.4 [13.5]	68.9 (10.4)
FTD	117	26.5 [43.6]	65.2 (9.0)
PSP	44	15.9 [40.9]	69.9 (7.3)
**Total**	**339**	**39.2 [27.4]**	**67.7** (**9.8**)
**Post-mortem brain tissue**			
**Sample Group**	**No. of individuals**	**Females** (**%**) **[% unknowns]**	**Average Chronological age** (**SD**)
**Cohort 2**			
Controls	8	62.5	75.8 (5.6)
FTLD-TDPA (*C9orf72*)	7	57.1	66.9 (4.8)
FTLD-TDPC (Sporadic)	8	50.0	72.9 (4.8)
**Total**	**23**	**56.5**	**72.0** (**6.1**)
**Cohort 3**			
Controls	14	64.3	78.4 (11.8)
FTLD-TDPA (GRN)	7	71.4	64.6 (7.6)
FTLD-TDPB (*C9orf72*)	13	61.5	63.8 (8.2)
FTLD-Tau (*MAPT*)	13	46.2	60.9 (7.6)
**Total**	**47**	**59.6**	**67.5** (**11.5**)
**Cohort 4**			
Controls	71	35.2	76.0 (8.0)
PSP	93	41.9	71.6 (5.3)
**Total**	**164**	**39.0**	**73.5** (**6.9**)

Cohort 1—purple; cohort 2—blue; cohort 3—green; cohort 4—yellow; FTD—frontotemporal dementia; PSP—progressive supranuclear palsy; FTLD—Frontotemporal lobar degeneration; FTLD-TDPA/B/C—FTLD with 43 kDa transactive response DNA-binding protein (TDP-43) positive inclusions, types A, B and C; *C9orf72*—*C9orf72* mutation carriers; *GRN*—*GRN* mutation carriers, FTLD-Tau—FTLD with tau-positive inclusions; *MAPT*—*MAPT* mutation carriers.

## Data Availability

The raw DNA methylation data from cohorts 1, 3, and 4 used in this study are openly available in Gene Expression Omnibus or EMBL-EBI ArrayExpress platform. Cohort 1: GEO accession number GSE53740 (https://www.ncbi.nlm.nih.gov/geo/query/acc.cgi?acc=GSE53740 (accessed on 1 December 2022)); Cohort 3: E-MTAB-12674 (https://www.ebi.ac.uk/biostudies/arrayexpress/studies/E-MTAB-12674?query=E-MTAB-12674 (accessed on 1 December 2022)). Cohort 4: GSE75704 (https://www.ncbi.nlm.nih.gov/geo/query/acc.cgi?acc=GSE75704 (accessed on 1 December 2022)). Additional data, including the raw data from cohort 2, is available on request from the corresponding author.

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
