# Peer review of "Epigenetic Age Acceleration in Frontotemporal Lobar Degeneration: A Comprehensive Analysis in the Blood and Brain"

_cells, 2023, doi:10.3390/cells12141922_

Round 1

Reviewer 1 Report

The authors present epigenetic analysis of peripheral blood and brain tissues of patients with Frontotemporal lobar degeneration (FTLD). The aim of the study is to determine the presence of accelerated epigenetic aging in FTLD compared to controls by calculating epigenetic age in both peripheral blood and brain tissues of multiple FTLD subtypes using several “DNA methylation clocks”.

From peripheral blood (cohort 1), 128 FTD patients, 44 progressive supranuclear palsy (PSP) patients and 193 healthy controls epigenome-wide DNA methylation profiles were extracted from public databases.

From brain studies, the authors enrolled three data sets: cohort 2 consisted of DNA methylation profiles generated from the frontal cortex grey matter from 16 individuals with FTD with TDP-43 pathology (8 C9orf72 mutation carriers and 8 sporadic cases) and 8 controls; cohort 3 consisted of DNA methylation profiles generated from the frontal lobe from 34 individuals with FTLD (7 GRN mutation carriers, 14 C9orf72 mutation carriers and 13 MAPT mutation carriers) and 14 controls; cohort 4 consisted of DNA methylation profiles generated from prefrontal lobe of 94 individuals with PSP and 72 controls extracted from public databases.

Age acceleration was calculated in peripheral blood dataset  using 4 clocks, namely DNAmClockMulti DNAmClockHannum PhenoAge and GrimAge For the post-mortem brain datasets DNAm age estimation was performed using 2 clocks, the pan-tissue DNAmClockMulti and the cortical tissue specific DNAmClockCortical

From all the comparisons, the authors found significant epigenetic age acceleration in the peripheral blood of both FTD and PSP patients compared to controls with DNAmClockHannum

The paper is well written and report innovative results applying new high-throughput technologies.

Nevertheless, all the paper is based in the application of DNA methylation clocks and further documentation about them should be provided in order to provide to the readers enough information to understand the results of the manuscript. I strongly suggest including in the introduction section a detail description of the basis of these clocks, what do they measure and which are the difference between them. Otherwise, it is difficult to understand the paper outcome

In this line my main concern is if all clocks measure age acceleration is difficult to justify why statistical significant results are only observed in blood using one out of four DNA clocks. In fact, the only significant result is presented in the sup table 1.

The age calculated for PSP by DNAmClockHannum is 73.9 whereas the chronological age is 69.9. However, the results obtained for the FTD cohort is 66.7 rather than 65.2. I am not sure the revelevance of this difference. Nevertheless, this result should be presented in the main manuscript.  Please add p-values and highlight in bold the statistical significant results.

All the other clocks showed lower ages compared to those chronological in blood. Similarly, in brain no significant results were observed and DNAmClockMulti which was used in both tissues showed also lower ages (cohort 3 and cohort 4) or similar ages (cohort 2). The results from DNAmClockCortical, specific for this tissue, show accelerating aging in all groups, but with the most remarkably difference in the control group.

Overall, the application of these DNA clocks failed to demonstrate an accelerating aging in FTD group and revealed an accelerating aging in PSP patients in blood using DNAmClockHannum and in brain tissue using  DNAmClockCortical, although no statistically significant.

In order to better understand the results presented, authors should modify results section focusing on statistical significant results, moving correlation results to an unique paragraph and avoid repetitive description such as the beginning of section 3.2

To verify if the age acceleration observed in peripheral blood of FTD and PSP could 233 be concordant in the post-mortem brain tissues, we performed epigenetic clock analysis using the pan-tissue DNAmClockMulti and the tissue specific DNAmClockCortical in 3 different cohorts constituting various FTLD subtypes. In particular, Cohort 2 included FTLD-236 TDP types A (C9orf72 mutation carriers) and C (sporadic cases) and controls, Cohort 3 included GRN (FTLD-TDPA), C9orf72 (FTLD-TDPB), and MAPT mutation carriers (FTLD-tau) and controls, and Cohort 4 included sporadic PSP cases (FTLD-tau) and controls.”

Minor comments:

- Numbers of individuals enrolled do not match between the material and methods section and the figure 1.

Cohort 1 is supposed to be formed by 128 FTD patients, 44 PSP patients and 193 healthy controls, a total simple size of 365 individuals. In figure 1the authors showed a reduction of controls and FTD sets, 178 controls rather than 193 and 117 FTD rather than 128. Similarly, for cohorts 2 and 3 there is one C9orf72 patient missing in both cohorts compared to those described in the material and methods. Finally, in cohort 4 there is one control and one PSP patient missing in the figure. Please revise or justify why these individuals have been excluded from the analysis.

- Figures and tables

Figures should be improved. Figure 2-5 are difficult to interpret in the way are currently presented. Author should assess to move them to a supplementary material.

Cohort description should be also move to main results.

Author Response

Reviewer #1

The authors present epigenetic analysis of peripheral blood and brain tissues of patients with Frontotemporal lobar degeneration (FTLD). The aim of the study is to determine the presence of accelerated epigenetic aging in FTLD compared to controls by calculating epigenetic age in both peripheral blood and brain tissues of multiple FTLD subtypes using several “DNA methylation clocks”.

 From peripheral blood (cohort 1), 128 FTD patients, 44 progressive supranuclear palsy (PSP) patients and 193 healthy controls epigenome-wide DNA methylation profiles were extracted from public databases.

From brain studies, the authors enrolled three data sets: cohort 2 consisted of DNA methylation profiles generated from the frontal cortex grey matter from 16 individuals with FTD with TDP-43 pathology (8 C9orf72 mutation carriers and 8 sporadic cases) and 8 controls; cohort 3 consisted of DNA methylation profiles generated from the frontal lobe from 34 individuals with FTLD (7 GRN mutation carriers, 14 C9orf72 mutation carriers and 13 MAPT mutation carriers) and 14 controls; cohort 4 consisted of DNA methylation profiles generated from prefrontal lobe of 94 individuals with PSP and 72 controls extracted from public databases.

 Age acceleration was calculated in peripheral blood dataset  using 4 clocks, namely DNAmClockMulti DNAmClockHannum PhenoAge and GrimAge For the post-mortem brain datasets DNAm age estimation was performed using 2 clocks, the pan-tissue DNAmClockMulti and the cortical tissue specific DNAmClockCortical

 From all the comparisons, the authors found significant epigenetic age acceleration in the peripheral blood of both FTD and PSP patients compared to controls with DNAmClockHannum

The paper is well written and report innovative results applying new high-throughput technologies.

Response: We thank the reviewer for the encouraging comments.

Nevertheless, all the paper is based in the application of DNA methylation clocks and further documentation about them should be provided in order to provide to the readers enough information to understand the results of the manuscript. I strongly suggest including in the introduction section a detail description of the basis of these clocks, what do they measure and which are the difference between them. Otherwise, it is difficult to understand the paper outcome

Response: We thank the reviewer for this suggestion and, in line with that, we have included a paragraph in the Introduction of the revised manuscript describing the nature of the clocks and their features (Lines 73­­­­­86).

In this line my main concern is if all clocks measure age acceleration is difficult to justify why statistical significant results are only observed in blood using one out of four DNA clocks. In fact, the only significant result is presented in the sup table 1.

Response: Although all clocks measure biological age, there is variability in the estimated DNAm ages because each clock was designed using different parameters, i.e., the DNAm values were obtained from different tissues and sample sizes, and different methodologies were to arrive at the final DNAm sites that are used as biomarkers to predict the epigenetic age. For example, DNAmClockMulti was designed using a large number of samples (n = 8000) that included 51 tissue and cell types, to generate a predictor of epigenetic age based on the DNAm at a final of 353 DNAm sites. Similarly, DNAmClockHannum makes use of 71 DNAm sites obtained using the blood tissue, and DNAmClockCortical uses 347 DNAm sites obtained from human cortical tissue to predict the epigenetic ages. The vast amount of heterogeneity involved while training the models results in the variability in DNAm ages predicted by each clock, and this is expected. We therefore used multiple clocks per type of tissue (i.e. blood and brain) and emphasize the results that are consistent across clocks (at least with effects in the same direction).

The age calculated for PSP by DNAmClockHannum is 73.9 whereas the chronological age is 69.9. However, the results obtained for the FTD cohort is 66.7 rather than 65.2. I am not sure the revelevance of this difference. Nevertheless, this result should be presented in the main manuscript.  Please add p-values and highlight in bold the statistical significant results.

Response: Although it is not completely clear to us what the reviewer’s point was here, we would like to emphasize that the comparisons we performed were between the disease cases versus controls. For example, regarding the DNAmClockHannum, the biological age of the PSP cases is estimated to be on average 4.2 years older than that of the corresponding controls [supplementary table 1, shows the age acceleration residuals after accounting for confounding factors, PSP (3.0) – controls (-1.2) = 4.2], and FTD 2 years older than controls [FTD (0.8) – controls (-1.2) = 2.0]. Figure 2a shows the same information in a more visual way and has the significant pairwise comparisons and corresponding p-values indicated at the bottom of the plots. Significant pairwise comparisons with the controls are now also highlighted in bold in supplementary table 1.

All the other clocks showed lower ages compared to those chronological in blood. Similarly, in brain no significant results were observed and DNAmClockMulti which was used in both tissues showed also lower ages (cohort 3 and cohort 4) or similar ages (cohort 2). The results from DNAmClockCortical, specific for this tissue, show accelerating aging in all groups, but with the most remarkably difference in the control group.

 Overall, the application of these DNA clocks failed to demonstrate an accelerating aging in FTD group and revealed an accelerating aging in PSP patients in blood using DNAmClockHannum and in brain tissue using  DNAmClockCortical, although no statistically significant.

In order to better understand the results presented, authors should modify results section focusing on statistical significant results, moving correlation results to an unique paragraph and avoid repetitive description such as the beginning of section 3.2

“To verify if the age acceleration observed in peripheral blood of FTD and PSP could 233 be concordant in the post-mortem brain tissues, we performed epigenetic clock analysis using the pan-tissue DNAmClockMulti and the tissue specific DNAmClockCortical in 3 different cohorts constituting various FTLD subtypes. In particular, Cohort 2 included FTLD-236 TDP types A (C9orf72 mutation carriers) and C (sporadic cases) and controls, Cohort 3 included GRN (FTLD-TDPA), C9orf72 (FTLD-TDPB), and MAPT mutation carriers (FTLD-tau) and controls, and Cohort 4 included sporadic PSP cases (FTLD-tau) and controls.”

Response: Thank you for this suggestion, we have moved the correlation results regarding comparisons between chronological age and DNAm age to a single paragraph at the beginning of the results to provide more context and better readability. The corresponding figures were moved to supplementary data in the revised manuscript (Lines 207225, Supplementary figures 1 and 2).

Minor comments:

- Numbers of individuals enrolled do not match between the material and methods section and the figure 1.

Cohort 1 is supposed to be formed by 128 FTD patients, 44 PSP patients and 193 healthy controls, a total simple size of 365 individuals. In figure 1the authors showed a reduction of controls and FTD sets, 178 controls rather than 193 and 117 FTD rather than 128. Similarly, for cohorts 2 and 3 there is one C9orf72 patient missing in both cohorts compared to those described in the material and methods. Finally, in cohort 4 there is one control and one PSP patient missing in the figure. Please revise or justify why these individuals have been excluded from the analysis.

Response: The sample numbers depicted in figure 1 were the final sample sizes obtained after excluding samples that did not pass quality control (QC). We have updated Figure 1 to make this clear and more details on sample sizes are also described in the methods and in table 1.

- Figures and tables

Figures should be improved. Figure 2-5 are difficult to interpret in the way are currently presented. Author should assess to move them to a supplementary material.

Response: In order to simplify the figures, we have moved the respective panels containing the correlation values, errors, and p-values for all cohorts (Figures 2 and 3) to the supplementary file in the revised manuscript (Supplementary figures 1 and 2). We have updated the x-axis labels in Figure 4 for easier interpretation. To simplify figure 5, we have moved the associations between age acceleration and disease onset to the supplementary data (Supplementary figure 3).

Cohort description should be also move to main results.

Response: We have now included Table 1 in the main manuscript with overall characterization of the cohorts.

Reviewer 2 Report

Authors have documented the outcomes of their research on FTLD disorders, highlighting the prospect of utilizing increased epigenetic age acceleration as a valuable biomarker. Their study holds notable significance in the field as it offers analyses for various cohorts and categories of disease. I would like to express my comment and pose some queries as follows. 

Are there any patients for which you had both brain and blood? Since there are different types correlations that are obtained through their data and it will be important to elaborate on this. 

I am curious, what authors can predict if they had the blood and brain of the same patients. Authors mention that this is the limitation of their study. Can they comment more on how they would predict those future studies based on current results considering the peripheral immune response in the blood. 

Can authors share the details about how many samples did not pass the quality control? 

Is it possible to represent the bar graphs with individual values so that we can actually see the distribution of the values?

Why is age acceleration in Controls in negative, how is the data normalized?

When the authors say, ‘‘In addition, concordant to previous reports, age acceleration was generally positively correlated with oligodendrocyte proportions and negatively correlated with neuronal proportions in brain tissue indicating the role of cellular proportions, which are typically altered in disease, towards epigenetic age estimations and validating our previous findings’’ Is there a way to normalize the data and then compare so that we can have a different insight in to the data? Perhaps within one category of the disease if there is a reduction in number of cells, can that be normalized to the area of the brain analyzed or any other approach?

It would be very informative if authors can add more information in figure 1 and indicate which tools were used at each the steps if applicable.
